# Long-Term Effect of Intra-Articular Adipose-Derived Stromal Vascular Fraction and Platelet-Rich Plasma in Dogs with Elbow Joint Disease—A Pilot Study

**DOI:** 10.3390/vetsci11070296

**Published:** 2024-07-01

**Authors:** Annika Bergström, Miriam Kjörk Granström, Lars Roepstorff, Mohammad J. Alipour, Kjerstin Pettersson, Ingrid Ljungvall

**Affiliations:** 1Department of Clinical Sciences, Faculty of Veterinary Medicine and Animal Science, Swedish University of Agricultural Sciences, P.O. Box 7070, 75007 Uppsala, Sweden; annika.bergstrom@anicura.se (A.B.); ingrid.ljungvall@slu.se (I.L.); 2University Animal Hospital, Swedish University of Agricultural Sciences, Ultunaallén 5A, P.O. Box 7070, 75007 Uppsala, Sweden; kjerstin.pettersson@slu.se; 3Department of Anatomy, Physiology and Biochemistry, Faculty of Veterinary Medicine and Animal Science, Swedish University of Agricultural Sciences, P.O. Box 7070, 75007 Uppsala, Sweden; lars.roepstorff@slu.se; 4Department of Veterinary Biosciences, Faculty of Veterinary Medicine, University of Helsinki, Agnes Sjöberginkatu 2, P.O. Box 66, 00014 Helsinki, Finland; mohammad.jaber.alipour@gmail.com

**Keywords:** dog, elbow dysplasia, OA, PRP, stem cell

## Abstract

**Simple Summary:**

Elbow osteoarthritis (OA) is a common cause of pain and lameness in dogs, often resulting from the developmental disorder elbow dysplasia. Currently, there is no effective treatment or cure for this disease. This study aimed to evaluate the effect of treating dogs with OA with stem cells (SVF, stromal vascular fraction) and blood plasma rich in platelets (PRP) derived from the dog’s own fat and blood, respectively. The mixture was administered as a single injection into affected elbows. Nineteen dogs with elbow OA were treated with SVF and PRP. Subjective and objective evaluations were performed before treatment, after six months, and after a minimum of one year. A “Symmetry Squares” graphic presentation of objective gait forces (peak force and impulse) was also used to compare changes in gait over time. The results showed that subjective evaluation of clinical lameness was improved at the six-month follow up evaluation and that the peak force was transferred from the hind limbs to the front limbs in the treated dogs after 12 months. However, the treatment failed to show a general evident effect. Further research should be conducted to evaluate whether SVF and PRP treatment should be recommended for dogs with elbow OA.

**Abstract:**

(1) Background: The aim of the current pilot study was to describe the long-term effects of a single intra-articular injection of autologous stromal vascular fraction (SVF) with platelet-rich plasma (PRP) in dogs with confirmed elbow OA, using orthopedic lameness scoring and kinetic and kinematic gait analysis. For comparison of normal long-term variation of gait over time, a group of healthy control dogs (CDs) was also evaluated. (2) Methods: A prospective longitudinal clinical pilot study investigating 19 client-owned dogs with elbow OA (OADs) treated with SVF and PRP and eight CDs not receiving treatment. The OAD and CD groups were evaluated before and after 6 and at least 12 months following treatment with SVF and PRP (OAD group) and twice with a six-month interval (CD group), respectively, through orthopedic examinations, goniometry, and kinetic and kinematic analyses (seven variables). (3) Results: The OAD had an increase in fore–hind peak force symmetry ≥12 months after treatment (*p* < 0.05), but no other objective variables changed over time. Orthopedic consensus scores had improved at ≥six months follow-up evaluation (*p* < 0.05). None of the investigated gait variables had changed at ≥six months follow-up evaluation in the CD group. (4) Conclusions: The current study could not confirm a significant benefit from SVF and PRP treatment in OADs, but future studies should be conducted in order to fully evaluate the potential of the treatment. The improvement seen in fore–hindlimb symmetry may represent an improvement in gait or an incidental finding.

## 1. Introduction

Elbow joint dysplasia (ED), or abnormal joint development, is the most common cause of elbow osteoarthritis (OA) in dogs. Medial coronoid disease and humeral osteochondrosis are two significant contributors to ED in dogs [1,2]. Ideally, treatment of ED and the resulting OA should heal cartilage defects and reverse the ongoing inflammatory process. However, the long-term prognosis of many medical and surgical therapies remains unclear [3,4]. Lameness and treatment effectiveness are frequently assessed using orthopedic lameness scoring, kinetic gait analysis, and owner questionnaires. While kinetic gait analysis is considered the gold standard, it is often more expensive and less readily available [5]. In contrast, lameness rating is widely available but considered less reliable due to the influence of subjectivity on the assessment [6]. Owner questionnaires are beneficial for repeated assessments over time [7,8,9]. Each method clearly has its inherent limitations. In human orthopedics, combined tests are used for lameness evaluation, for the purpose of achieving more accurate results.

Mesenchymal stem cell therapy is an alternative or complementary treatment to NSAIDs, surgery, and physiotherapy [10]. Adipose tissue in dogs contains multipotent stem cells of mesenchymal origin [11], which can be used to retrieve mesenchymal stem cells [12,13]. The use of a stromal vascular fraction (SVF) allows harvesting with minimal manipulation, reducing the requirement for in vitro expansion compared with a culture of mesenchymal stem cells [14]. Commercial methods for harvesting these cells in an SVF are available, enabling a one-step surgical procedure [15,16]. Clinical trials and case series in humans have reported decreased pain and improved cartilage growth following SVF application [17]. In dogs, several clinical studies have shown positive short-term outcomes (two to six months) after autogenic SVF joint treatment. However, evaluation methods vary, with some studies using force plates for objective measurement and double-blinded evaluations, and others relying solely on subjective lameness scores [18,19,20,21,22]. Allogenic stem cells have also been reported for treatment of elbow dysplasia in dogs [10].

Mesenchymal stem cells with capability to develop into cartilage tissue have been harvested from adipose tissue in dogs [23]. Instead of repair with fibrocartilage, hyaline cartilage may be formed in vivo after mesenchymal stem cell treatment. The exact mechanism of action in vivo remains unclear but has been suggested to be due to the excretory effect from stem cells and direct differentiation [24]. Immune modulation and downgrading the inflammatory response including joint osteoarthritis have been reported in humans [25,26,27], and the mechanisms in dogs appear to be similar [28,29].

The possibility to administer stem cells intravenously (IV) may be an alternative or used in combination with intraarticular injections. The effect of IV infusion of mesenchymal stem cells in dogs with elbow ED is still unclear, as reported by Olsen et al. [30].

Platelet-rich plasma (PRP) contains significant amounts of growth factors and cytokines, making it the most accessible and affordable source of autologous cytokines and growth factors [31]. The use of PRP has been reported to inhibit the deleterious effects of inflammation and to counteract the development of OA, with a chondroprotective effect in vivo in humans [32,33]. Platelet-rich plasma has been reported to improve lameness scores, reduce pain, and increase peak vertical forces in dogs with OA at follow-up examinations performed after three months [34]. When PRP is activated, it may function as a scaffold for an SVF [35]. A double-blinded, placebo-controlled trial based on owner questionnaires reported improvements in dogs with hip dysplasia 24 weeks after receiving SVF and PRP treatment [16]. However, evidence regarding long-term effects of SVF and PRP treatment in dogs with elbow OA is lacking.

The aim of the current pilot study was to describe the long-term effects of a single intra-articular injection of autologous SVF with PRP in dogs with confirmed elbow OA (OADs), using orthopedic lameness scoring and kinetic and kinematic gait analysis. For comparison of normal long-term variation of gait over time, a group of healthy control dogs (CDs) was also evaluated.

## 2. Materials and Methods

The study was performed at the University Animal Hospital at the University of Agricultural Sciences, Uppsala, Sweden.

Inclusion criteria: Dogs with elbow OA (OADs) and healthy, non-lame control dogs (CDs) were included in the study. Included OADs had to have a confirmed diagnose of unilateral or bilateral elbow OA based on diagnostic imaging (radiography or CT) findings, evaluated by a board-certified radiologist (ECVDI) or a veterinarian under their supervised training. To be eligible for enrollment, the OAD were required to previously have undergone NSAID treatment and rehabilitation including physical therapy exercises, with insufficient results—meaning continued pain from elbows and persistent lameness despite conservative treatment. Before or at the time of SVF/PRP therapy, OADs could have received arthroscopic treatment; however, this was not a requirement for participation.

Clinically healthy client-owned dogs without history or clinical signs of systemic diseases, lameness, or joint disease were included in the CD group. All dogs included in the current study were also included in a study by Kjörk-Granström et al. [36].

Exclusion criteria for the OADs included signs of systemic disease based on clinical examination and hematological/biochemical blood profiles, or ongoing lameness not related to OA, such as due to an infectious process or an acute trauma. Within both groups, dogs were excluded if they were receiving medical treatments, including immunosuppressive medications that potentially could mask pain responses (e.g., corticosteroids) but recent NSAID treatment for OADs under certain circumstances did not preclude inclusion. Treatment with NSAIDs was prohibited for the OAD group from 48 h prior to and until two weeks following the SVF and PRP injection but was recorded and permitted as needed thereafter. Treatment with NSAIDs was registered (name of substance, dosage, duration of treatment), and the decision to initiate or stop ongoing therapy was decided by the responsible veterinarian (AB, MK). Due to long-term follow-up, not being able to treat with NSAIDs, if considered necessary, was considered unethical for the OAD group. Any pain medication, including NSAIDs, was an exclusion criterion for the CD group.

Clinical experimentation was approved by the Uppsala Ethics Committee (C102/15). Prior to participation, all owners gave their written consent.

Clinical evaluation at inclusion:

On the day of inclusion for all dogs, a complete health check was performed including physical evaluation. For the OADs, hematological and biochemical blood profiles were evaluated to assure health and exclude signs of systemic disease. Additionally, elbow radiographs were taken, unless diagnostic imaging (CT or radiographs) had been performed less than six months prior to enrolment.

Orthopedic evaluations:

Dogs were independently examined by a board-certified orthopedic surgeon (AB) and a physical therapist (KP) at inclusion, before any potential treatment strategies had been implemented (all dogs), after six months (all dogs), and at least 12 months (OADs only) after enrolment into the study. A scoring system for clinical orthopedic evaluations was used as described by Kjörk Granström et al. [36]. Lameness at a walk and trot (graded 0–5), joint pain, palpable passive range of motion (PROM), goniometric and muscle measurements [37] for awake lateral recumbent dogs were included in the scoring. Finally, when all data had been collected, but before results from the objective gait analysis were available, both examiners discussed their individual scorings and agreed on a common orthopedic consensus score, in one of three categories; (1) normal, (2) mildly or (3) moderately to severely affected (Table 1).

Only treated OAD joints are included in the table. Values are reported as mean and standard deviation. Values with * differed significantly between the OAD and CD groups at the first examination. Values with ^#^ differed significantly between data obtained before treatment and after six months after treatment (*p* < 0.05) for OAD. Within each row of continuous data, values with the same superscript letter did not differ significantly (*p* > 0.05) between OADs and CDs. The unilateral healthy limb was not evaluated in the orthopedic evaluations. NSAID = non-steroid anti-inflammatory drug, BCS = body condition score, ROM = range of motion, PF = peak force, Imp = impulse, Sin = sinister, Dx = dexter.

Kinetic and kinematic evaluations:

Kinetic and kinematic measurements were performed at the same intervals as the orthopedic evaluations. The data collection and processing used were identical to those described by Kjörk Granström et al. (Appendix A) [36].

Gait velocity was measured in m/s (total peak force (PF); symmetry (Sym); fore–hind PF Sym, fore PF Sym, total impulse (I) Sym, fore–hind I Sym, fore I Sym, and kinematic range of motion (ROM) of the elbow). All the kinetic variables were normalized for body weight. Speed was normalized to mean speed at each session for each dog. Based on PF and I, the distribution of forces between all four legs was presented for each dog in a “Symmetry Squares” (SS) graph (Appendix B). Diagonal forces represent symmetries between contralateral fore and hind limbs. Blinded evaluations of the SS graphs were reviewed independently by three clinicians to evaluate any change in symmetry forces (lameness detection) and any change over time, at the same intervals as the orthopedic evaluations for the CD and OA group. All occasions for each individual dog were presented in the same graph and graded as unchanged, improved, or deteriorated over time (Section 3, Figure 1).

PRP preparation:

Before induction of anesthesia, 16–18 mL of venous blood was collected from each OAD under strictly aseptic conditions. The autologous PRP was thereafter prepared with a validated method following the *MediVets In-Clinic Procedure Manual* (Footnote I).

Preparation of SVF:

General anesthesia was induced with propofol (Propovet Multidose, Zoetis, Finland), and isoflurane (IsoFlu, Zoetis, Finland) was used for maintenance. Adipose tissue (approximately 20 g) from the thoracic area was collected surgically using standard surgical procedures in the three first dogs included in the study. Due to seroma formation in two of the three dogs, abdominal falciform fat was collected from the remaining dogs without complications thereafter. Samples were processed in-house, using a commercial kit to yield SVF pellets according to the manufacturer’s instruction (*Medi Vet America Manual*, Footnote I). The final cell suspension with the ASC pellet and 2 mL of PRP was placed into a photobiostimulation unit (Medivet Stem Cell LED Activator ML-1) for 20 min before injection into the elbow joint under strict aseptic conditions. The adipose-derived cell population was activated by irradiating the cells with the ML-1, which generated certain wavelengths of light in the visible light spectrum to stimulate growth and differentiation of cells (Appendix C). In bilaterally affected dogs; both elbows were treated with SVP and PRP.

Cell counts of SVF and PRP:

Total nucleated cell count per gram of adipose tissue was estimated for all dogs via laser flow cytometry of the final SVF solution (Footnote II) by diluting 0.1 mL cell suspension with 0.4 mL normal saline in an empty vacutainer blood collection tube and placing it in the analyzer.

Statistical evaluations:

Statistical analyses were performed using a commercially available software program (Footnote III). Group data are presented as mean and standard deviation (SD). Data were analyzed using descriptive as well as inferential statistics. Potential differences in baseline continuous-outcome biomechanical variables (listed in Table 1) between the OAD and CD groups, as well as potential differences in the clinical orthopedic consensus score between the first and the six-month follow-up examinations (for both the OADs and the CDs) were investigated using the Wilcoxon rank sum test. For the continuous-outcome biomechanical variables listed in Table 1, differences between time-points of examinations (before treatment, after 6 and at least 12 months, respectively, after treatment for the OAD group, and six months apart for the CD group) were investigated separately for the OAD and CD groups using a mixed (repeated) linear model including dog identity as a random variable and time-point of examination. The normality of the distributions of model residuals were ensured by visual inspection. The Tukey HSD test was used as a post hoc test for pairwise comparisons of variables found to be significant in the overall analysis. A value of *p* ≤ 0.05 was considered significant for the analyses.

## 3. Results

### 3.1. Dogs

The OAD group consisted of 19 dogs and included the following breeds: Belgian Shepherd (Malinois) (n = 1), Bernese Mountain dog (n = 2), German Shepherd (n = 3), German Spaniel (n = 1), Labrador Retriever (n = 7), Leonberger (n = 1), mixed breeds (n = 2), Rottweiler (n = 1), and Bullmastiff (n = 1). The CD group consisted of eight dogs and included the following breeds: German Shepherd (n = 1), German Wirehaired Pointer (n = 1), Labrador Retriever (n = 3), mixed breed (n = 1), Rottweiler (n = 1), and Smooth Collie (n = 1). Bodyweight was significantly lower in the CD group (Table 1).

Seventeen OADs had radiographic findings of OA including osteophytes, two of them with visible ED lesions (osteochondrosis of medial humeral condyle). In two other dogs, ED (medial coronoid disease) with OA was confirmed through CT scan. Five OADs had unilateral elbow OA and 14 had bilateral elbow OA disease. The unilateral elbow disease was left-sided in two dogs and right-sided in three dogs. See Table 1 for further information regarding the demographic data of included dogs.

Fifteen dogs were arthroscopically treated in one or both elbows before stem cell therapy. Three dogs had their second elbow arthroscopy and stem cell therapy on the same anesthetic occasion. The contralateral elbow had earlier been arthroscopically treated in these three dogs. In one OAD, no arthroscopy had been performed. All dogs treated arthroscopically had fragment removal from medial coronoid disease and/or medial humeral condyle osteochondrosis and debridement of necrotic tissue.

Two dogs (10%) developed short-term complications from the treated elbows after treatment, both resolving spontaneously within three to four days; one developed a local swelling in the treated elbow and one showed pain from the elbow.

### 3.2. Treatment Response and Cell Counts of SVF

Table 1 presents NSAID treatment, orthopedic lameness scoring, orthopedic consensus score, kinetic gait analysis, and kinematic range of motion for OADs and CDs. Before treatment, 67% of the dogs were treated with NSAIDs; this was reduced to 26% after six months and 37% after at least 12 months. Six months after stem cell therapy, the following NSAIDs were administered as needed, based on a responsible veterinarian’s (AB or MKJ) decision: meloxicam (n = 2), firocoxib (n = 2), and carprofen (n = 1).

Nineteen dogs were evaluated after 6 months and eleven (58%) after at least 12 months. Mean time from treatment to the at-least 12-months follow-up was 1.5 years ± 0.56. Three owners indicated long distance to the animal hospital as the cause for not participating in the last follow-up, one dog was euthanized due to OA, one due to lymphoma, and one dog did not participate due to chronic kidney disease. One dog was excluded due to a shoulder joint trauma and one dog due to lumbar pain that developed after six months follow-up. The clinical orthopedic consensus score for the right and left fore limbs had changed six months after treatment in the OAD groups (*p* < 0.05 for both the right and the left fore limb) (Table 1). Due to loss of 8/19 OADs at the at-least 12-months follow-up, statistical comparison was not performed for the categorical data for this examination occasion. Before treatment, total PF, total I, and fore–hind PF and I symmetry forces were lower in the OAD group compared with the CD group; fore–hind PF symmetry in the OAD group increased at the follow-up after at least 12 months compared with before treatment (*p* = 0.007) (Table 1).

Results from the blinded evaluation of any changes in symmetry forces (SS graphs) over time are presented in Table 1. The eleven remaining OADs did not move between categories (worse/unchanged or improved) between follow-up after six and at least 12 months, meaning that if they were improved at six months, they had maintained that improvement after at least 12 months. The CDs were assessed as unchanged in 7/8 dogs after six months. One dog was considered to have put on more weight in the front limbs at six months. The SS graphs of four individual dogs are presented as examples in Figure 1, and the fore–hind PF Sym index over time in all dogs is graphically presented in Figure 2. Dogs in the OAD group had a wide variety of negative numbers while the CD group was more homogenous. However, a clear overlap was seen in fore–hind PF Sym between the CD and OAD groups. The variation in change over time in the OAS group should be noted; e.g., OA10 showed deterioration, ogs OA1 and OA16 showed no apparent changes, whereas OA6, OA7, and OA8 improved.

The cell count of nucleated cells in the canine SVF was 8.9 × 10^6^ (±3.13 × 10^6^) cells/gram of adipose tissue.

## 4. Discussion

This study showed improvements in clinical orthopedic consensus score six months after treatment and in fore–hind PF symmetry in the 11 OADs remaining in the study at least 12 months after treatment. Other variables did not change significantly after six or after at least 12 months. Accordingly, in this pilot study, the overall clear long-term efficacy of SVF/PRF treatment could not be confirmed in dogs with elbow OA.

The ambiguous response to treatment may have several causes, such as no or limited therapeutic effect, only short-term benefits, or undetectable changes due to the small sample size. Additionally, the SVF procedure technique may have influenced the results. For instance, SVF/PRP injection following an arthroscopic procedure may not be ideal due to the risk of dilution and/or extravasation. Recently, Blumhagen et al. reported the effect of elbow arthroscopy in dogs immediately after intra-articular injections. The authors concluded that the arthroscopy procedure per se did not result in greater extravasation of fluid. However, the injected volume of fluid was of importance for the amount of extravasation [38]. These findings reflected the fact that the dogs that received SVF and PRP with a total volume less than 2 mL IA during the same anesthetic procedure as an arthroscopy had a low percentage of extravasation.

Treatment with SVF and PRP could potentially improve quality of life by slowing down disease progression. The absence of deterioration during the year of follow-up could potentially indicate a clinical effect of the treatment as OA overall is a progressive disease. In addition, in the study by Kjörk Granström [36], dogs with fore-limb OA had a significantly greater hind peak force than dogs without the disease. The improvement in fore–hind limb symmetry following OADs’ treatment in the current study may possibly therefore correspond to an improvement in gait or represent an incidental finding. A higher percentage of OADs returning for the scheduled follow-up examinations in addition to enrollment of a clinical control group for comparison would have been required to fully evaluate these findings.

This study reports a longer follow-up time compared with several other stem cell studies in dogs, but the number of dogs included in the present study was low. Due to the loss of 8/19 OADs at follow-up after at least 12 months, categorical data such as orthopedic consensus score were evaluated only six months after treatment. Of the eight missing dogs, only one was reported to be excluded due to elbow OA (OAD11), but it is likely that the true number of missing dogs with clinical OA pain was higher.

Vilar et al. reported a short-term (three months) positive effect after treatment with cultured stem cells without PRP in dogs with hip OA in terms of objective gait analysis, before relapse to levels before treatment at 180 days after treatment [22]. Clinical use of SVF or cultured cells has been sparsely studied but has been reported in one study by Marx et al., where SVF was reported to show more clearly positive clinical results compared with cultured cells [21]. The possible reasons for this remain unknown, but SVF also contains other cells, which may be important for the healing process in the OA joint.

In human research, PRP has been found to enhance stem cell therapy [39], which may also apply for dogs. In a blinded study including 39 dogs evaluated with subjective lameness scoring after treatment, Cuervo et al. [20] reported that PRP may increase the duration of clinical effect with SVF. In contrast to our study, they did not include objective gait analysis. The combination of SVF and PRP as a local injection to the joint and as an IV infusion may enhance the therapeutic effect and should be further studied in the future.

Scoring of lameness and gait analysis measurements in dogs with mild and/or bilateral lameness is of lower magnitude, making it harder to evaluate changes over time. As seen in Figure 2, some individuals from the OAD group had similar fore–hind PF Sym as CDs, which could have been clinically identified as normal variation in gait. Vilar et al. reported an increase in ground reaction forces (GRFs) in the more severely lame limb and a decrease in GRFs in the less lame limb after stem cell therapy [22]. The lameness scores presented in Table 1 include eight dogs with lameness grade 4/5 before treatment. The severity of lameness scores decreased over time, and after six and at least 12 months follow-up, no dog showed 4/5 in lameness score. However, due to the small sample size and loss of eight dogs at follow up after at least 12 months, the lameness grading is reported only descriptively.

The SS evaluation also identified an increase in forces in one or two fore limbs in 47% of the OADs after six months, in 63% of the OADs after at least 12 months, and in 12% (one dog) of the CDs after six months. A bias not related to treatment could have resulted in the decrease of lameness severity over time. Bilateral lameness is considered more difficult to evaluate. The SS graphs will be useful for the physician as a tool to graphically discern how forces are distributed and change over time, as seen in the current investigation (Figure 1). It was not possible to detect a difference in outcomes after treatment between uni- or bilaterally lame dogs. Elbow OA is considered a chronic disease; it may wax and wane but generally worsens over time, with an increase of lameness and NSAID therapy. Before SVF and PRP treatment, all treated dogs had been periodically treated with NSAIDs and rehabilitation, but the effect was considered clinically insufficient according to owners and veterinarians. The usage of NSAIDs decreased from 67% before treatment to 26% after six months. After at least 12 months, only 11 dogs were included and at that time; 37% received NSAIDs. It is possible that SVF and PRP therapy may reduce the need for NSAIDs in dogs, but further research, preferably in the form of double-blinded studies, is needed to verify this finding as the owners might be biased in regard to the treatment.

BCS reported in Table 1 was higher for the OADs compared with the CDs; 90% of the OADs scored 6/9 or higher at the last follow-up, while approximately 50% scored 6/9 or higher in the CD group. Weight reduction has been reported to improve movability in OA dogs [40], and the increased BCS during the study time may have negatively affected the OADs.

Based on the gait analysis results presented in Figure 2, some of the young, severely lame dogs treated with SVF/PRP (OAD 6, 7, 8, and 14) may have benefited more from the therapy compared with older and severely lame dogs (OAD 3, 9, and 10). Such a potential difference in treatment effect between young and old dogs should be further investigated. The influence of age on the activity and number of human stem cells has been discussed [33]. Furthermore, the chronicity of joint disease in older dogs may have had an influence on the SVF and PRP treatment effect.

Limitations: Significant limitations of this pilot study were the relatively low number of individuals included and the loss of 8/19 OADs at follow-up after at least 12 months, which may have affected the results. Other limitations were the lack of a placebo-treated OAD group, and the fact that the clinical evaluations were not blinded to treatment group or control group. This may have caused a risk of bias towards clinical improvement in the treated OADs. All the orthopedic evaluations were also subjective and prone to bias [6]. One inclusion criterion for OADs was insufficient response to conservative treatment, and the inconsistent severity of OA among included dogs may have affected the results.

In this pilot study, the number of administered active stem cells was undefined. Fresh SVF is heterogeneous with different types of cells and may be difficult to evaluate accurately due to debris. However, Maki et al. reported no difference in effect when three different concentrations of adipose mesenchymal stem cells were administered [41].

Adding diagnostic imaging or arthroscopy of treated elbows to the follow-up examination protocol would clearly have been valuable as an additional tool for objective evaluation of therapy. For financial reasons, this was not included. Finally, making age a criterion for inclusion would have provided more information about how age affected the efficacy of treatment.

## 5. Conclusions

The current study could not confirm a clear benefit from SVF/PRP treatment in OADs. The improvement in fore–hind symmetry seen at least 12 months after treatment in the OAD group may represent an improvement in gait or an incidental finding. Further studies should be conducted, including a placebo control group for comparison and a higher number of individuals, to fully evaluate the potential of SVF/PRP treatment.

Footnote:I.Provided by Medivet Biologics LLC, Lidcombe, Australia. Medivet is currently Arden Animal Health.II.IDEXX LaserCyte, Westbrook, ME, USA.III.JMP 16.0, Cary, NC, USA

## Figures and Tables

**Figure 1 vetsci-11-00296-f001:**
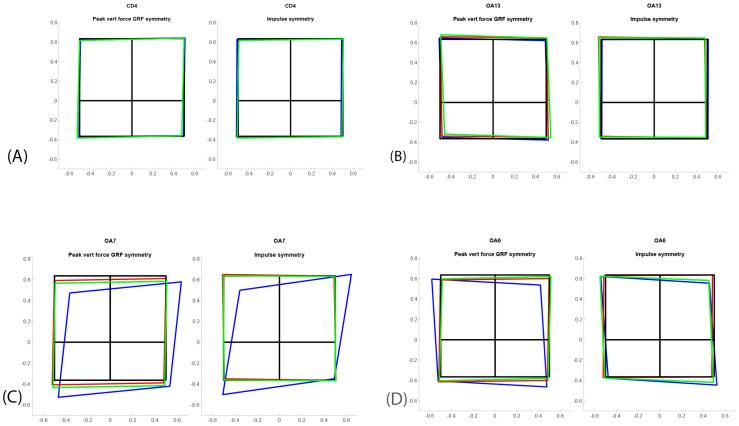
Symmetry squares for four different dogs: (**A**) normal dog (CD4); (**B**) low-grade OA dog (OA13); (**C**) left forelimb OA and visible lameness (OA7) improving in symmetry after treatment; (**D**) right forelimb OA and visible lameness (OA6) improving in symmetry over time. The squares represent left fore limb, right fore limb, left hind limb, and right hind limb as if the dog were being viewed from above. The lame dog transfers the weight from the painful limb, which can be observed in OA6 and OA7, but not in OA13. Blue color = before treatment/first evaluation, green color = six months, red color ≥ 12 months.

**Figure 2 vetsci-11-00296-f002:**
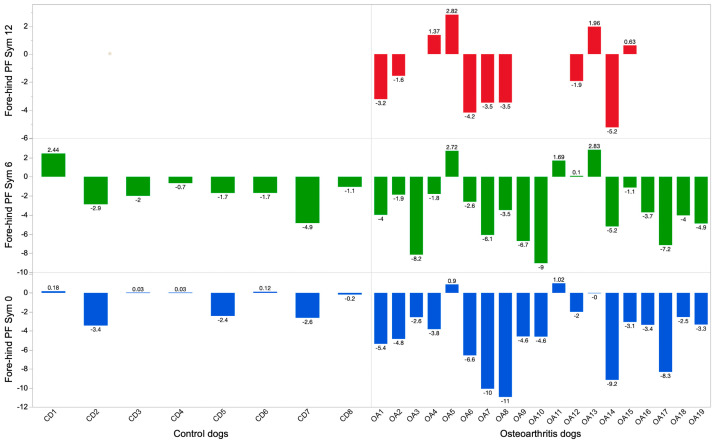
Fore–hind peak force symmetry (PF Sym) in non-lame dogs (CDs) (**left**) and dogs with elbow osteoarthritis (OADs) (**right**). Blue color = before treatment/first evaluation, green color = 6 months, red color ≥ 12 months for the OAD group.

**Table 1 vetsci-11-00296-t001:** Demographic data, orthopedic lameness scoring, kinetic gait analysis, and kinematic range of motion results in 19 dogs with osteoarthritis (OADs) investigated before treatment and after six and at least 12 months after stromal vascular fraction and platelet-rich plasma treatment had been performed, and in eight non-lame dogs (CDs) investigated six months apart.

	OADs at Inclusion before Treatment (n = 19)	OADs 6 Months (n = 19)	OADs ≥ 12 Months (n = 11)	CDs at Inclusion (n = 8)	CDs 6 Months (n = 8)
Age (years)	4.2 ± 2.9	-	-	5.8 ± 2.8	-
Sex (female/male)	10/9	10/9	7/4	4/4	4/4
Bodyweight (kg)	35.3 ± 9.1	35.7 ± 9.7	33 ± 9.0	27.0 ± 6.6 *	27.6 ± 6.6
BCS: U/N/OU: 1–3 under weight N: 4–5 normal weight O: ≥6 overweight	0/5/14	0/4/15	0/1/10	0/4/4	0/4/4
NSAIDs (Yes/No), NSAID% Descriptive only	12/7 (63%)	5/14 (26%)	3/8 (37%)	0/8	0/8
Orthopedic evaluations					
Lameness score (trot), presented with number of dogs in each lameness grade 0–5: (0/1/2/3/4/5)	Dx 2/3/5/2/5/0 Sin 2/4/3/4/3/0	D 8/3/3/3/0/0 Sin 5/9/0/2/0/0	Dx 2/6/1/2/0/0 Sin 4/2/3/0/0/0	All scored 0	All scored 0
Orthopedic consensus score (normal/mild/moderate and severe)	Dx 3/8/8 Sin 4/9/6	Dx 10/4/5 ^#^ Sin 11/6/2 ^#^	Dx 3/6/2 Sin 7/3/1	Dx 8/0/0 Sin 8/0/0	Dx 8/0/0 Sin 8/0/0
Measured passive ROM elbow (degrees)	Dx 115.6 ± 11.3 ^a^ Sin 120.2 ± 7.3 ^a^	Dx 16 ± 10.6 ^a^ Sin 119 ± 7.8 ^a^	Dx 115 ± 10.5 ^a^ Sin 119 ± 10.7 ^a^	Dx 125.6 ± 2.9 ^a^ Sin 125.4 ± 3.7 ^a^	Dx 124.9 ± 2.5 ^a^ Sin 126.0 ± 2.5 ^a^
Measured muscle mass (cm)	Dx 28.7 ± 2.5 ^a^ Sin 29.5 ± 2.9 ^a^	Dx 29.1 ± 2.5 ^a^ Sin 9.6 ± 2.6 ^a^	Dx 28.7 ± 3.0 ^a^ Sin 29.1 ± 2.8 ^a^	Dx 26.6 ± 2.1 ^a^ Sin 27.0 ± 1.98 ^a^	Dx 27.6 ± 2.1 ^a^ Sin 28.3 ± 2.1 ^a^
Kinetic and kinematic evaluations					
Total PF symmetry	−7.1 ± 4.92 ^a^*	−5.65 ± 3.8 ^a^	−3.99 ± 1.74 ^a^	−1.63 ± 1.58 ^a^*	−2.98 ± 1.13 ^a^
Fore–hind PF symmetry	−4.38 ± 3.44 ^a^*	−3.29 ± 3.48 ^ab^	−1.48 ± 2.75 ^b^	−1.04 ± 1.51 ^a^*	−1.55 ± 2.06 ^a^
Fore PF symmetry	0.52 ± 5.36 ^a^	0.20 ± 3.3 ^a^	−0.28 ±1.25 ^a^	0.24 ± 0.46 ^a^	0.07 ± 0.32 ^a^
Total Imp symmetry	−5.17 ± 3.72 ^a^*	−4.27 ± 3.26 ^a^	−2.55 ± 1.38 ^a^	−1.62 ± 0.61 ^a^*	−1.82 ± 1.12 ^a^
Fore–hind Imp symmetry	−1.71 ± 2.67 ^a*^	−0.43 ± 2.86 ^a^	−0.19 ± 1.70 ^a^	0.55 ± 0.92 ^a^*	0.38 ± 1.42 ^a^
Fore Imp symmetry	0.55 ± 5.58 ^a^	−1.04 ± 3.94 ^a^	−0.76 ± 1.73 ^a^	−0.05 ± 0.90 ^a^	0.68 ± 0.86 ^a^
ROM Dx at walk	57.8 ± 6.1 ^a^	55.6 ± 8.5 ^a^	56.9 ± 6.6 ^a^	54.5 ± 6.3 ^a^	53.1 ± 5.1 ^a^
ROM Sin at walk	58.2 ± 1.7 ^a^	56.6 ± 9.2 ^a^	58.3 ± 2.1 ^a^	57.5 ± 2.7 ^a^	54.8 ± 7.0 ^a^
Symmetry squares blinded evaluation: worse/unchanged/improved	Baseline (n = 19)	5/5/9 (n = 19)	1/3/7 (n = 11)	Baseline (n = 8)	0/7/1

Only treated OAD joints are included in the table. Values are reported as mean and standard deviation. Values with * differed significantly between the OAD and CD groups at the first examination. Values with ^#^ differed significantly between data obtained before treatment and after six months after treatment (*p* < 0.05) for OAD. Within each row of continuous data, values with the same superscript letter did not differ significantly (*p* > 0.05) between OADs and CDs. The unilateral healthy limb was not evaluated in the orthopedic evaluations. NSAID = non-steroid anti-inflammatory drug, BCS = body condition score, ROM = range of motion, PF = peak force, Imp = impulse, Sin = sinister, Dx = dexter.

## Data Availability

The datasets used and analyzed during the current study are available from the corresponding author on reasonable request. Data are not public due to GDPR and privacy restrictions.

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
