# Peer review of "Long-Term Effect of Intra-Articular Adipose-Derived Stromal Vascular Fraction and Platelet-Rich Plasma in Dogs with Elbow Joint Disease—A Pilot Study"

_vetsci, 2024, doi:10.3390/vetsci11070296_

Round 1

Reviewer 1 Report

Comments and Suggestions for Authors

Long-term effect of intraarticular adipose-derived stromal vascular fraction and platelet-rich plasma in dogs with elbow joint disease – a pilot study

Dear authors, thank you for the submissiono f this excellently written paper. Very informative. I only have a couple comments.

Line 301 Please reference this paper and discuss its implications in your paper. 

 Impact of arthroscopy on post-procedure intra-articular elbow injections: A cadaveric study Emalee M. Blumhagen DVM | Daniel I. Spector DVM, DACVS (Small Animal) | Anthony J. Fischetti DVM, MS, DACVR May 2024 veterinary surgery

Line 497  586  Please present in a better way. It seems unorganized.

Author Response

Dear reviewer 1,

thank you for the valuable comments you made on our pilot study. We feel that adjustments now made has improved the manuscript. Please find the point-by- point response in the attachment. 

Sincerely, Miriam Kjörk Granström 

Reviewer 2 Report

Comments and Suggestions for Authors

Thank you for submitting this interesting paper to Veterinary Science. My opinion is as follows.

Introduction

1.     Please add details of previous reports on the efficacy of SVF+PRP in OA dogs due to ED.

2.     Please explain how this study differs from previous SVF reports.

Materials and methods

1.     Please state the reason for using 8 CD dogs.

Results

1.     Line 287; “6” is not shown with superscript.

2.     Figure2; The explanation shown in the upper left corner is duplicative.

Discussion

Please add past evidences (e.g. in vitro) on the therapeutic effect of SVF.

Discussions should be structured by content.

1.     Please discuss methods (in vivo and in vitro) to confirm that adipose-derived cells in SVF administered into the elbow joint are acting at the lesion site.

2.     Please add references showing that adipose-derived cells in the administered SVF were involved in lesion repair.

3.     Please discuss why SVF+PRP administration did not provide a significant benefit in this study.

4.     Consider ways to enhance the therapeutic effect of SVF+PRP. Is intravenous administration better than local administration? And why?

5.     Please explain the mechanism by which the efficacy of SVF+PRP seen in this study was obtained.

6.     Please explain why you did not compare them to untreated OA dogs.

7.     Since this is a pilot study, please explain how you plan to approach the next step to address the issues you have identified in this study.

Author Response

Dear reviewer 2, 

thank you for your valuable comments made on our pilot study. We feel that the manuscript now is improved. Please see attached document for our point-by-point response. 

Sincerely, Miriam Kjörk Granström 

Reviewer 3 Report

Comments and Suggestions for Authors

Comments and suggestions to the authors

Thank you for submitting to Animals this interesting article that attempts to describe the long-term effects of a single intra-articular injection of autologous stromal vascular fraction (SVF) with platelet-rich plasma (PRP) in dogs with elbow OA by means of an orthopaedic lameness score and kinetic and kinematic gait analysis. 

Here are my comments.

Inclusion/exclusion criteria could be improved.

"Insufficient results" referring to NSAID treatment should be explained. 

In both groups, dogs receiving medical treatments that might mask the pain response were excluded, but which medical treatments are the authors referring to? Many NSAIDs have varying degrees of analgesic activity. 

Have the requirements for NSAID treatment been catalogued and standardised? The name of the substance, dosage, duration of treatment should be given in the text.

I think it would be useful to standardise more the age of the subjects included in the study. The age of the subjects ranged from a minimum of 1.3 years to a maximum of 7.1 years, including both young and adult subjects, a wide range within which the development of a chronic, progressive osteoarticular disease can vary considerably from year to year. 

Similarly, the use of a homogeneous sample by selecting subjects of the same race would have made it possible to standardise the weight of the subjects. In addition, information on the subjects' body condition (BCS) shows an over-representation of overweight subjects, which appears to be a key factor in the progression of osteoarthritis. This finding should be taken into account in the discussion.

With regard to the treatments administered in the experimental group, I think the study design should be clarified, especially the explanation of the number of subjects who underwent arthroscopy.

From the text, it appears that of the 19 subjects included in the study, 15 subjects had an arthroscopic procedure in one or both elbows prior to SVF/PRP therapy, and 3 bilaterally affected subjects had an arthroscopic procedure in the first limb prior to SVF/PRP therapy and the same procedure in the other limb at the same time. Arthroscopy is therefore an additional procedure to SVF/PRP in some subjects and concomitant in others. The inclusion of both classes of subjects in one category, and the lack of standardisation of the performance of the arthroscopic procedure and the administration of SVF/PRP therapy, may constitute a BIAS for inclusion.

Finally, the lack of imaging to objectively assess the progression of osteoarthritis during the post-therapy observation period does not allow for an objective evaluation of the therapy, which does not appear to be effective in the long term based on the results presented.

Comments on the Quality of English Language

A linguistic revision is required.

Author Response

Please, see attached 

Reviewer 4 Report

Comments and Suggestions for Authors

This study aims to describe long-term effects of a single intraarticular injection of autologous stromal vascular fraction with platelet rich plasma in dogs with confirmed elbow OA, using orthopedic lameness scoring and kinetic and kinematic gait analysis. The experimental design of this paper is reasonable, the language expression is appropriate, and it has clinical application significance. But the description of the conclusions needs to be further refined. The difference between results and conclusions should be distinguished when revising. The "Simple summary" could be shorter and simpler, because it for general readers instead of just for professional. 

Abstract

1.       The first sentence of the conclusion section of the abstract is result, not the conclusion. Please give a more detailed description of the conclusion.

Keywords

2.       Keywords do not need to be numbered.

Introduction

3.       The advantages and disadvantages of complementing different assessment methods can be considered. (Line 67-69)

Materials and Methods

4.       The formatting of paragraphs should be reviewed.

5.       The inclusion condition was inconsistent OA severity in dogs that did not respond to NSAIDs treatment, which may have affected the results. Please add a note of limitations of the study. (Line 119-114)

6.       The source of the PRP should be stated in this section. (Line 159-162)

Result

7.       Whether the number is expressed in Arabic or English should be the same, please check the full text and modify it. (Line 264)

8.       The author should distinguish between the notes (figure legends) and the results in the manuscript and revise them. (Figure 2)

9.       3.13 x 106 should be 3.13 x 106.(Line 287)

Conclusions

10.    The conclusion of the article needs to be summarized again, as the significance of this article cannot be highlighted by the current conclusion. (Line 377-381)

References

11.    The format of the references needs to be re-examined, some DOI are missing or incomplete.

Author Response

Please, see attached

Round 2

Reviewer 2 Report

Comments and Suggestions for Authors

Thank you for your reply. However, one answer was insufficient.

Although a larger number of control dogs may be beneficial in the evaluation, the number of dogs used should be reduced to the minimum necessary for the use of experimental animals. In your study plan, how many animals are needed for the control group and for the placebo group that you plan to implement in the future? You still have not answered why you chose 8 dogs for the control group.

Author Response

Dear reviewer 2, 

thank you for your valuable comments and points. Please see the attached file, I hope that this will provide answers for the remaining questions. 

Sincerely, Miriam Kjörk Granström 

Reviewer 3 Report

Comments and Suggestions for Authors

Comments and suggestions for the authors
I have no further suggestions. Well done.

Author Response

Dear reviewer 3, 

Thank you so much for your comments and suggestions. We feel that adding the recent and highly relevant article you suggested really improved our manuscript. 

Sincerely, Miriam Kjörk Granström